# Effects of Maximal and Submaximal Anaerobic and Aerobic Running on Subsequent Change-of-Direction Speed Performance among Police Students

**DOI:** 10.3390/biology11050767

**Published:** 2022-05-18

**Authors:** Nenad Koropanovski, Robin M. Orr, Milivoj Dopsaj, Katie M. Heinrich, J. Jay Dawes, Filip Kukic

**Affiliations:** 1Department of Criminalistics, University of Criminal Investigation and Police Studies, 11080 Belgrade, Serbia; nenad.koropanovski@kpu.edu.rs; 2Tactical Research Unit, Bond University, Gold Coast, QLD 4229, Australia; rorr@bond.edu.au; 3Faculty of Sport and Physical Education, University of Belgrade, 11030 Belgrade, Serbia; milivoj.dopsaj@fsfv.bg.ac.rs; 4Department of Kinesiology, Kansas State University, Manhattan, KS 66506, USA; kmhphd@ksu.edu; 5School of Kinesiology, Applied Health and Recreation, Oklahoma State University, Stillwater, OK 74074, USA; jay.dawes@okstate.edu; 6Police Sports Education Center, Abu Dhabi Police, Abu Dhabi 253, United Arab Emirates

**Keywords:** agility, metabolic fitness, tactical fitness, athletic performance

## Abstract

**Simple Summary:**

Change-of-direction maneuvers are frequently performed by police officers and athletes. These maneuvers are typically performed with the intention of being maximally fast. Often, an officer or an athlete will run at a certain pace before commencing a change-of-direction speed maneuver. Depending on the duration and intensity of this running activity, their performance of the change-of-direction speed maneuver may be reduced. This study determined the degree to which the preceding maximal and submaximal anaerobic and aerobic activity affect the subsequent performance of the change-of-direction speed maneuver. We found that both anaerobic and aerobic running activities decreased the speed of the subsequent performance on the Illinois Agility Test. We also found that anaerobic running at 85% and 90% had a greater impact on change-of-direction speed performance than did aerobic running at these intensities. Above 90% intensity, anaerobic and aerobic performance similarly impacted the change-of-direction speed. As such, given the requirement for tactical personnel and intermittent, multidirectional sports athletes to perform a change-of-direction speed maneuver following a period of submaximal anaerobic or aerobic activity, increasing fitness may be a means of reducing the negative impacts of preceding submaximal impacts on change-of-direction speed performance.

**Abstract:**

Change-of-direction speed (CODS) directly impacts success in sports, police, and military performance. Movements requiring CODS are often preceded by aerobic or anaerobic running. Therefore, this study investigated the effects of maximal and submaximal anaerobic and aerobic running on subsequent CODS performance. A sample of 50 police students (42% female and 58% male) performed a maximal 300-yard shuttle run test (SR300y) and a 2.4-km Cooper test (CT2.4km) at maximal effort and also at 95, 90, 85, 80, and 75% of maximal effort. CODS was assessed using the Illinois Agility Test (IAT) immediately following each intensity level of each test at 12 separate testing sessions. To avoid fatigue, the period between each consecutive session was a minimum of 3 days. Paired samples *t*-tests were used to determine the differences between the two conditions (anaerobic lactic and aerobic) and for the IAT. A repeated measure analysis of variance with a Bonferroni post hoc test was used to analyze partial effects of different running intensities on the IAT. A significant reduction in speed was observed between the initial IATmax time and the IATmax time after performing the SR300y at intensities of 95, 90, 85, and 80% of maximal speed on this test. IAT performance was significantly slower when performed after the CT2.4km at 95 and 90% of maximal aerobic speed. The effects of the SR300y on the IAT were significantly greater than the effects of the CT2.4km. No significant differences were found by sex. Building up to 90% intensity, anaerobic running has a greater negative impact on subsequent CODS performance than does aerobic running.

## 1. Introduction

Change-of-direction speed (CODS) has been identified as a performance ability needed for a variety of sports, as well as for many physically demanding occupations, such as law enforcement, military, and firefighting [1,2,3,4]. CODS is the non-perceptual component of agility, or the ability to change direction rapidly in a pre-planned manner [1,2,5]. It includes repeated bouts of acceleration, sprinting, deceleration, and changes of direction [1,2]. Besides technique and sprinting speed, CODS performance depends on muscle qualities, such as strength, reactive strength, and power [2,6,7,8], each of which is supported by neuro-physiological characteristics and adaptations [9,10,11,12].

Considering the mechanical characteristics and physiological background of CODS performance, its importance is clear in multidirectional sports (e.g., soccer, football, handball, basketball) [13,14,15] or in physically demanding occupations [3,16,17]. Effective performance in these sports and occupations also depends on other important physical abilities, such as aerobic and muscular endurance or metabolic power [18,19]. This is true because these activities and tasks typically last for a prolonged time, repeatedly employing CODS. A basketball player, for example, requires optimal CODS capabilities throughout an entire game [20], while a soccer player performs 150–250 brief, intense actions during a match [21]. In an occupational setting, a police officer may need to chase a suspect who then attempts multiple changes in direction [22]. For example, in a study of 48 police officers wearing occupational loads (9.55 ± 0.95 kg), Joseph et al. [23] found a 23% decrease in speed when officers switched from a 20 m linear sprint to a figure-eight course. The authors postulated that the decrease in CODS was initiated by the greater relative loads worn by the officer. Therefore, it is critical to understand the extent to which previous activities may affect CODS.

Indeed, numerous tests have been used to assess CODS performance and the anaerobic and aerobic running capabilities in athletes and tactical athletes, and the maximal 300-yard shuttle run, 2.4-km Cooper running test, and the Illinois Agility Test are among those that are often used [18,24,25,26,27]. All three are field tests, are easy to conduct repeatedly, and require minimum equipment, while in return, they provide a valid estimation of anaerobic power, aerobic power, and CODS performance efficacy [26,28,29]. Considering the complexity of athletes’ training processes, their competition schedules, and rest periods, the choice of assessment procedures typically considers the aforementioned criteria. These tests can be used as part of the training session and thus used occasionally to track the athletes’ physical preparedness or the effects of a particular training cycles (i.e., micro, meso, macro). Similarly, tactical athletes (e.g., police), while required to be physically ready, also must balance work and training schedules, court visits, and family obligations [30]. Therefore, it is of importance to integrate assessment procedures into their training schedules whenever possible. This can often be attained by choosing appropriate tests such as the 300-yard shuttle run, 2.4-km Cooper test, and Illinois Agility Test.

The potential relationship of CODS with other physical abilities has been explained in detail, but there is limited information on how preceding activities may affect CODS. The effects of submaximal to maximal anaerobic and aerobic running tasks on subsequent CODS performance are unknown. Therefore, the aim of this study was to investigate the effects of different anaerobic and aerobic running intensities on subsequent CODS performance. We hypothesized that different anaerobic and aerobic running intensities preceding the CODS task would have a significant impact on CODS performance.

## 2. Materials and Methods

### 2.1. Study Design and Procedures

Data were collected from April to June 2021 across 12 testing sessions. During the first two testing sessions (Testing Days I and II), anthropometric data were collected, and participants were familiarized with the testing protocol. In addition, before each testing session, participants performed a 10 min warm-up consisting of running variations and calisthenics exercises. Testing sessions consisted of several testing stations that participants completed consecutively in the following order:DAY III: (1) Illinois Agility Test, (2) 5 min rest, (3) 300-yard shuttle run test;DAY IV: (1) Illinois Agility Test, (2) 5 min rest, (3) 2.4 km Cooper test (maximal effort), (4) Illinois Agility Test (no rest between 3 and 4);DAYS V, VI, and VII: as per DAY IV above, but with the 2.4 km Cooper test performed at 95, 90, and 85% of maximal aerobic speed;DAY VIII: (1) Illinois Agility Test, (2) 5 min rest, (3) 300-yard shuttle run with 95% intensity, (4) Illinois Agility Test (no rest between 3 and 4);DAYS IX, X, XI, and XII: as per DAY VIII above, but with the 300-yard shuttle run performed at 90, 85, 80, and 75% of maximal effort, (4) Illinois Agility Test (no rest between 3 and 4).

### 2.2. Participants

Consulting the G*power analysis (see Statistics, Section 2.6, for details) for the minimal sample size, we initially started the study with 57 participants, but 3 female and 4 male students dropped out during the study. One student had to leave the campus and miss several testing sessions due to a death in the family, four students missed several testing sessions due to sickness, and two students left the study due to lack of motivation. Therefore, the final sample consisted of 50 students (42% female). The main characteristics for males were age = 19.8 ± 0.5 years, body height = 182 ± 0.1 cm, body weight = 82.3 ± 9.3 kg, and body mass index = 24.8 ± 2.5 kg/m^2^. The main characteristics for female participants were age = 19.7 ± 0.5 years, body height = 171 ± 0.1 cm, body weight = 64.1 ± 5.4 kg, and body mass index = 22.0 ± 1.8 kg/m^2^. All participants passed the physical fitness requirements for recruitment and were attending physical education classes (self-defense, use of force, and strength and conditioning) three times per week. They were informed about the aim of the study, testing protocol, and potential risks. Including criteria were that the respondent was completely healthy and wanted to participate in the study. The exclusion criterion was the failure of the respondent to perform all tests. Only participants who signed informed consent were included in the study. This study received ethical approval (440-2).

### 2.3. Settings

This study was conducted at the University of Criminal Investigation and Police Studies, Belgrade, Serbia. Participants were accommodated at the university’s campus, which allowed for the control of their exercise activities between the testing sessions and the rest periods between the sessions. During the testing sessions, they did not have any other training activities. Therefore, they were sufficiently rested and had the same conditions. More specifically, the rest period between sessions was 3 days, i.e., 72 h. The tests were conducted at the same time of day, in same footwear and apparel, and under identical environmental circumstances (temperature 21 ± 2.1 °C, relative humidity 58%, barometric pressure 1001 ± 9 mb). Anaerobic endurance and CODS were assessed in a nonslip sports hall, while aerobic endurance was assessed on an outdoor, circular track with a length of 200 m (Figure 1). The distances between the finish line of the anaerobic endurance test, aerobic endurance test, and the starting line of CODS were 5 m and 10 m, respectively. After the anaerobic and aerobic tests, participants walked swiftly to the CODS test starting line.

### 2.4. Measurement Procedures

#### 2.4.1. Anaerobic Running

Maximal and submaximal anaerobic running were assessed using the 300-yard shuttle run test (SR300y). The test includes 12 lengths of a 25-yard distance. It has been previously used with sports and tactical populations [27,28,31,32]. The test–retest reliability of this test (ICC = 0.83) has been determined elsewhere [33]. Individual times obtained from a maximal SR300y (SR300ymax) were used to calculate intensities of 95, 90, 85, 80, and 75% (SR300y95, SR300y90, SR300y85, SR300y80, and SR300y75, respectively) for each participant in order to cover the range of anaerobic intensities (i.e., BLa > 4.0 mmol/L) [34,35,36]. For example, the equation for calculating the time for an intensity of 95% was:Time for SR300y95 = time for SR300ymax + (time for SR300ymax × 0.05)(1)

The running speeds were governed by a visual light pacer (Pacer2, KulzerTEC, Santa Maria da Feira, Portugal), which consisted of a 22.86 m (25 y) long line equipped with 26 LED lights per meter, from 0 to 22.86 m points. The LED lights flashed sequentially to signify the tempo at which participants were required to maintain in order to keep up with the preset speed. The pacer was stationed alongside the running track.

Each participant’s heart rate (HR) was monitored using a Polar (V800, Polar Electro, OY; Kempele, Finland) device secured with a chest strap. Maximal HR (Hrmax) was recorded at the end of each test to control for running intensity. Blood lactate levels (BLa) were collected 3 minutes after the test when BLa accumulation has been reported to be at its highest [37]. BLa was measured only after the SR300ymax so as to ensure that participants provided maximal effort as all other SR300y intensities were calculated according to SR300ymax.

#### 2.4.2. Aerobic Running

Maximal and submaximal aerobic running were assessed using the 2.4 km Cooper test (CT2.4km), which has previously been used in law enforcement recruits [38,39]. The reliability (ICC = 0.99) and accuracy (bias correction = 0.994 for distance and 0.956 for HR) of the Cooper running test were shown to be acceptable in long-distance runners [40]. The validity and reliability were further confirmed in university students [29] and healthy adults [41]. The time obtained to complete the CT2.4km as quickly as possible (i.e., maximal effort) was used to calculate the running intensities of 95, 90, and 85% maximal aerobic speed. For example, to calculate maximal aerobic speed at 95% intensity, the following equation was used:Time for CT2.4km95 = time for CT2.4km + (time for CT2.4km × 0.05)(2)

Further, the obtained result in seconds was divided by 24 to calculate the 100 m pace. Introduced ranges were used to determine the effects of different aerobic intensities on subsequent CODS performance. As the participants were well-familiarized with the Cooper test, they were instructed to keep the pace as constant as possible during the maximum performance. The same instruction was given when they ran at set intensities. Two experienced instructors were placed at every 100 m and provided feedback with clear instructions (e.g., keep up the pace, accelerate slightly, or slow down slightly) to ensure that participants ran at a set intensity. In addition, HRmax was measured immediately after the maximal CT2.4km to ensure that participants ran maximally. The mean HRmax was 194.43 ± 11.33 beats per minute for females and 196.83 ± 8.57 beats per minute for males.

#### 2.4.3. Change-of-Direction Speed

CODS was assessed using the standardized Illinois Agility Test (IAT) following the procedures that have been previously reported in the literature [3,16]. The time to complete the test was measured by timing gates (Fitro Light Gates, Fitronic, Bratislava, Slovakia) and expressed in seconds, with a precision of 0.01 s. The test was performed two times, and the better result was recorded for the analysis. The IAT was also repeated on DAY XIII for reliability purposes and to control for learning and training effects, given the number of testing sessions. Test–retest reliability was high (ICC = 0.985), which was similar to that (ICC = 0.96) reported by Hahana et al. [26]. A paired samples *t*-test for test–retest differences was not statistically significant.

### 2.5. Variables

Differences between the pre- and post-running tests were calculated and expressed in percentages. The following variables were computed for the IAT: ∆IATmax—difference in IAT after CT2.4km run at maximal intensity; ∆IAT95%—difference in the IAT after SR300y and CT2.4km run at 95% intensity; ∆IAT90%—difference in the IAT after SR300y and CT2.4km run at 90% intensity; ∆IAT85%—difference in the IAT after SR300y and CT2.4km run at 85% intensity; ∆IAT80%—difference in the IAT after SR300y run at 80% intensity; and ∆IAT75%—difference in the IAT after SR300y run at 75% intensity. Participants did not perform the IAT after the maximal SR300y as they were not able to do so due to fatigue.

### 2.6. Statistics

The analyses were performed using JASP Team 2021 (Version 0.16) and SPSS (Version 23.0., Armonk, NY: IBM Corp) statistical software. Descriptive statistics for both sexes were calculated for the mean, standard deviation (SD), minimum (Min), and maximum (Max) values. The Shapiro–Wilk test was used to assess the normality of distribution. Independent samples *t*-tests and the Mann–Whitney test were used to determine between-sex differences. Paired samples *t*-tests and Wilcoxon signed-rank were used to determine the differences between the two conditions (anaerobic lactic and aerobic) for the IAT. A repeated measures analysis of variance (ANOVA) with the Bonferroni post hoc test was used to analyze the partial effects of different running intensities on the IAT. In instances where the distribution violated conditions of normality, results were adjusted according to the Friedman test and the Conover post hoc test. Significance was set at *p* < 0.05. Cohen’s d calculations were used to determine the effect size of each running intensity in both conditions [42]. The effect sizes were defined as small (d = 0.2–0.49), medium (d = 0.5–0.79), large (d = 0.8–1.29), and very large (d = 1.3 and larger) [43]. We used G*Power to determine the minimum sample sufficient to detect the large effect size. For the independent sample *t*-test, 52 participants (26 in each group) were recommended; for the paired sample *t*-test, 15 participants were recommended; and for the repeated measures ANOVA, 10 participants were recommended.

## 3. Results

Male students outperformed female students on the IAT, SR300y and CT2.4km (*p* < 0.001) and produced significantly higher (*p* = 0.02) BLa levels, while maximal HRs were similar for both sexes (Table 1).

The IAT performance was significantly (*p* = 0.01) slower after the SR300y at 90% compared to the IAT after CT2.4km at 90% (Table 2). The repeated measure ANOVA determined significant difference between the IATmax and the IAT performed after the SR300Y at 95, 90, 95, and 80%, as well as the difference between the IATmax and the IAT after the CT2.4km at maximal, 95, 90, and 85%.

Table 3 shows the results of the *t*-test between the relative effects (%) attained by SR300y and CT2.4km at different intensities. The effect of SR300y performed at 90 and 85% on the IAT was significantly larger compared to the same intensities of running for the CT2.4km. The lowest intensities in SR300y and CT2.4km were insufficient to produce significant negative effects on IAT performance.

Post hoc analysis revealed gradually lower effects of anaerobic and aerobic running on IAT performance as participants ran the SR300y and CT2.4km at lower intensities (Figure 2). The largest relative difference occurred after participants ran the CT2.4km at maximal intensity and after the SR300y and CT2.4km at 95% of maximal aerobic speed. The lowest relative difference occurred after the lowest running intensities. Sex did not play a significant role (*p* = 0.20–0.80), although small effect sizes occurred after the SR300y 80% (d = 0.3) and the CT2.4km 95 and 90% (d = 0.4 and 0.3, respectively).

## 4. Discussion

This study investigated the effects of different anaerobic and aerobic running intensities performed immediately prior to a CODS task on subsequent CODS performance. The main findings revealed that anaerobic and aerobic running had negative effects on subsequent CODS performance. Anaerobic running at 85 and 90% of maximum negatively impacted CODS performance to a greater extent than did aerobic running at the same intensity, while intensities over 90% hindered CODS performance equally, whether performed anaerobically or aerobically. These effects were relatively consistent among both sexes with no significant between-sex differences in CODS performance after each running intensity. Given that both the anaerobic endurance SR300y and the IAT were short-duration, explosive movements, it was not surprising that taxing the anaerobic system using a SR300y produced greater impacts on CODS. The CT2.4km, while still utilizing some anaerobic metabolism at higher intensities, is predominantly aerobic in nature. Thus, as participants performed at 90% maximal aerobic speed during the CT2.4km, it is not surprising that the impact on CODS was commensurate with their performance following the SR300y.

For law enforcement officers, these outcomes raise concerns. The anaerobic activities characterized by high intensity (e.g., chase) can precede a CODS requirement (e.g., an offender begins weaving and changing direction to escape). Furthermore, these impacts may be exacerbated by the occupational loads officers are required to carry. A study by Orr et al. [3] found that the inclusion of 10 kg of occupational load reduced CODS performance (IAT unloaded = ~23.17 s, IAT loaded = ~24.14 s, *p* < 0.001). These authors also found that correlations between fitness measures (standing long jump, sit ups, and push-ups) and CODS performance increased once occupational loads were added. The relationship between general fitness and CODS performance is supported by Pihlainen et al. [44]. The authors reported that 3000 m running performance was the second strongest predictor of a military simulated task consisting of rushes with changing directions, crawling, sprinting, lifting, and carrying while soldiers were fully loaded. Subsequently, it is not surprising that a study conducted with Dutch military recruits showed that implementing agility training improved CODS performance and reduced injury rates in this population [45]. Based on the above, the fitness of individuals can be expected to influence CODS performance, whereby the importance of metabolic fitness is amplified when there is a need for preceding anaerobic and aerobic tasks. Therefore, highlighting the development of the ability to interchangeably utilize running at different intensities and CODS may be crucial for successful and safe job performance.

Similarly, in sports games such as handball, soccer, and basketball, athletes spend most of their playing time at lower intensities, with game-changing CODS performances only constituting a small part of the game [21,46,47]. Póvoas et al. [46] reported that handball athletes spent around 80% of the game time standing still or walking, while only 0.4% of the time could be assigned to sprinting. Mohr et al. [48] reported that standing, walking, or jogging constitute most of the time in a soccer game (i.e., about 78% in top- and about 82% in moderate-class players). Furthermore, top-class players spent significantly higher amounts of time sprinting compared to moderate-class players (1.4% vs. 0.9% of the match time). These top-class players were also found to perform better in the Yo-Yo intermittent recovery test, suggesting that they were less fatigued by activities preceding their maximal sprint efforts [48]. Similar patterns were observed in basketball, where the majority of the time, players performed at low to moderate intensities, with lower proportions of total playing time spent in high intensity activities [47,49]. As such, while explosive CODS movements may be performed infrequently, they typically occur immediately following a period of anaerobic and aerobic running activities. To that end, our study design, by mimicking this scenario, determined the aerobic and anaerobic running intensities after which the CODS performance decreased and to what extent the preceding running type and intensity affected the subsequent CODS performance. This information has clear implications for planning and programming training processes for law enforcement officers.

### 4.1. Strengths and Limitations

The presented study has several strengths. First, the study was designed to mimic the natural order of events in sports and law enforcement as well as in military tasks where the task performance effectiveness depends on CODS performance. Thus, the presented findings can be generically applied to different tactical occupation situations. It is important to emphasize that the participants were under strict and controlled conditions during the entire study period. Last, but not least, the controlled intensities used in this study approach have provided a richness of data regarding the effect of preceding anaerobic and aerobic running on CODS performance.

The study also has some limitations. The sample consisted of police cadets, a population with a good general level of physical fitness and with a relatively narrow age range. For that reason, the effects of aging could not be investigated. Finally, the study was focused on the time fluctuation of CODS performance after physical stress, and any psychological stress was neither introduced nor considered.

### 4.2. Clinical Application of Findings

For tactical personnel required to perform intermittently explosive and multidirectional movements, the higher the intensity of the preceding movement, the greater the decrement on CODS. This is of note, given that these CODS movements can occur at critical times during which performance must be optimal. Thus, not only would tactical occupations benefit from CODS performance testing that mimics requirements (i.e., preceded by submaximal/maximal) anaerobic or aerobic performance, but these personnel should focus on improving their fitness to optimize CODS performance by mitigating the fatiguing impacts from immediately preceding activities.

## 5. Conclusions

Building up to 90% intensity, anaerobic explosive movements have a greater impact on subsequent CODS performance as compared to aerobic movements although the latter still negatively impact CODS performance. Above 90% intensity, anaerobic and aerobic performance both negatively impact CODS. As such, given the requirement for tactical personnel and intermittent multidirectional sports athletes to perform CODS movements following a period of submaximal anaerobic or aerobic activity, the means to mitigate this impact are of importance. Increasing fitness may be a means of reducing the negative impacts of preceding submaximal impacts on CODS performance; however, further research is needed.

## Figures and Tables

**Figure 1 biology-11-00767-f001:**
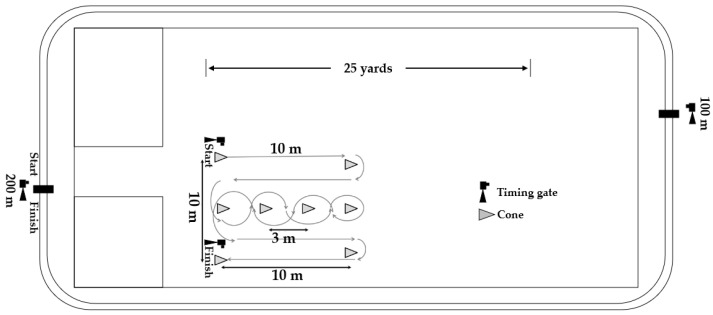
Schematic of test positioning.

**Figure 2 biology-11-00767-f002:**
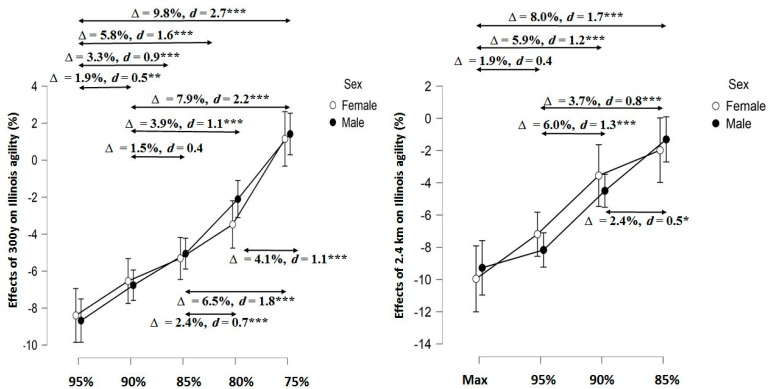
Effects of different running intensities on the Illinois Agility Test. * Significant at *p* < 0.05, ** Significant at *p* < 0.01, and *** Significant at *p* < 0.001, d—Cohen’s effect size, ∆—relative difference between effects obtained after two different running intensities. Max, 95%, 90%, 85%, 80%, 75%—running intensities of SR300y and CT2.4km.

**Table 1 biology-11-00767-t001:** Descriptive characteristics of the study participants and differences between sexes.

Variables	Females (*n* = 21)				Males (*n* = 29)			
	Mean	SD	Min.	Max.	Mean	SD	Min.	Max.
IAT (s) **	18.569	1.537	17.869	23.308	17.677	1.139	15.914	19.842
SR300y (s) **	73.80	5.86	65.92	94.58	62.57	3.40	57.22	70.51
La3min *	13.25	1.75	9.80	16.10	14.38	1.57	9.60	16.50
HRmax300y (b/min)	188.10	8.80	170.00	202.00	189.86	8.13	176.00	204.00
CT2.4km (s) **	750.95	68.23	662.00	921.00	634.14	73.00	527.00	924.00
HRmax2.4km (b/min)	194.43	11.33	168.00	215.00	196.83	8.57	181.00	219.00

Note: * Significant at *p* < 0.05. ** Significant at *p* < 0.01. IAT—Illinois Agility Test, SR300y—Shuttle run 300 y, La3min—Lactate level after 3 min of recovery, HRmax300y—Maximal heart rate after SR300y, CT2.4km—Cooper test 2.4 km, HRmax2.4km—Maximal heart rate after CT2.4km.

**Table 2 biology-11-00767-t002:** Descriptive characteristics, repeated measure ANOVA, and paired sample *t*-tests for the Illinois Agility Test.

Variables	Post SR300y		Post CT2.4km	
	Mean	SD	Mean	SD
IATPostMax (s)	-	-	20.373 *	1.866
IATPost95% (s)	19.778 *	3.344	20.031 *	1.652
IATPost90% (s) ^$^	19.822 *	1.723	19.294 *	1.637
IATPost85% (s)	19.532 *	1.655	18.867	1.512
IATPost80% (s)	19.008 *	1.691	-	-
IATPost75% (s)	18.266	1.499	-	-

Note: ^$^ Significant difference between the IAT performed after SR300y and CT2.4km, * Significantly different at *p* < 0.05 compared to initial IAT values. IATPostMax—Illinois Agility Test performed after maximal SR300y and CT2.4km. SR300y—shuttle run 300 y, CT2.4km—Cooper test 2.4 km. IATPost95%, IATPost90%, IATPost85%, IATPost80%, and IATPost75%—Illinois Agility Test performed after SR300y at intensities of 95, 90, 85, 80, and 75% or after CT2.4km at 95, 90, and 85%.

**Table 3 biology-11-00767-t003:** Differences in effects obtained by SR300y and CT2.4km.

Variables	Post SR300y		Post CT2.4km		*t*-Test between Conditions	
	Mean Difference	SD	Mean Difference	SD	Mean Difference	d
∆IATmax			9.7	4.3		
∆IAT95%	6.5	15.7	7.9	3.4	0.7	0.16
∆IAT90%	6.7	3.5	4.1	3.5	2.6 ***	0.56
∆IAT85%	5.2	3.4	1.53	3.9	3.6 ***	0.91
∆IAT80%	2.4	2.9	-	-	-	-
∆IAT75%	1.5	3.4	-	-	-	-

Note: *** Significant at *p* < 0.001. IATPostMax—Illinois Agility Test performed after maximal 2.4 km Cooper test, IATPost95%, IATPost90%, IATPost85%, IATPost80%, and IATPost75%—Illinois Agility Test performed after SR300y at intensities of 95, 90, 85, 80, and 75% or CT2.4km at 95, 90, and 85%. ∆IAT95%, ∆IAT90%, ∆IAT85%, ∆IAT80%, and ∆IAT75%—relative difference between the initial IAT time and time obtained after the SR300y at intensities of 95, 90, 85, 80, and 75%, or the CT2.4km at 95, 90, and 85%, expressed in %.

## Data Availability

Data is available upon request from filip.kukic@gmail.com or nenad.koropanovski@kpu.edu.rs.

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
