# Peer review of "Effects of Maximal and Submaximal Anaerobic and Aerobic Running on Subsequent Change-of-Direction Speed Performance among Police Students"

_biology, 2022, doi:10.3390/biology11050767_

Round 1

Reviewer 1 Report

Dear authors, I attach my comments.

Simple Summary: 
This section has not been included.

Abstract
The abstract adequately summarises the research, including the objective, methodology, main findings of the results and conclusions. 
Also include the percentage of males (42% female).

Keywords:
Adequate.

Introduction
Scarce.
Further work is needed on why these tests are used and not others.

Materials and Methods
2.1. Study design and procedures

Do men and women do the same? Same tests?
No previous studies cited. 

2.2. Participants

Males and females cannot be mixed.

A sample consisted of 50 students (42% female). The main sample characteristics 86 were age = 19.7 ± 0.5 years, body height = 177 ± 8 cm, body weight 74.7 ± 12.0 kg, and body mass index = 23.6 ± 2.6 kg/m2.

Need to indicate at which point in the training sessions the tests were performed.

Include for all tests and assessments the reference authors and their validity and reliability in measuring what they say they measure. This will be of great help to young researchers.

Results 

They are adequately presented.

Discussion 
In the discussion all statements should have previous research supporting or contradicting the results obtained. This should be reviewed. Many citations are missing.

Conclusions
They are clear.

References

Some minor flaws. Numbers together without separation. Use of inappropriate formatting in the year, number and pages.

Author Response

Simple Summary: This section has not been included.

Response: Simple summary is added now.

This is added:

Simple summary: Change of direction maneuvers are frequently performed by police officers and athletes. These maneuvers are typically performed with the intention to be maximally fast. Often, an officer or an athlete runs at a certain pace before commencing to change of direction speed maneuvers. Depending on the duration and intensity of this running activity, their performance of change of direction speed maneuver may be reduced. This study determined to what degree the preceding maximal and submaximal anaerobic and aerobic activity affect the subsequent performance of the change of direction speed maneuver. We found that both anaerobic and aerobic running activities decreased the speed of subsequent performance of the Illinois Agility Test. We also found that the anaerobic running at 85 and 90% had a greater impact on the change of direction speed performance than the aerobic running at these intensities. Above 90% intensity, anaerobic and aerobic performance both similarly impact the change of direction speed. As such, given the requirement for tactical personnel and intermittent multidirectional sports athletes to perform a change of direction speed maneuvers following a period of submaximal anaerobic or aerobic activity, increasing fitness may be a means of reducing the negative impacts of preceding submaximal impacts on change of direction speed performance.

Abstract The abstract adequately summarises the research, including the objective, methodology, main findings of the results and conclusions. Also include the percentage of males (42% female).

Response: Thank you. We added percentage for males as well.

Keywords: Adequate.

Response: Thank you.

Introduction:  Scarce. Further work is needed on why these tests are used and not others.

Response: We added one paragraph to explain why these tests were used.

This is added:

“Indeed, numerous tests have been used to assess CODS performance, anaerobic and aerobic running capabilities in athletes and tactical athletes, thereby the maximal 300-yards shuttle run, 2.4 km Cooper running test, and Illinois agility test are among those that are often used [1–5]. All three are field tests, are easy to conduct repeatedly, and require minimum equipment, while in return providing a valid estimation of anaerobic power, aerobic power, and CODS performance efficacy [4,6,7]. Considering the complexity of athletes’ training processes, their competition schedules and rest periods, the choice of assessment procedures typically considers the aforementioned criteria. These tests can be used as part of the training session and thus used occasionally to track the athletes’ physical preparedness or effects of particular training cycles (i.e., micro, meso, macro). Similarly, tactical athletes (e.g., police) while required to be physically ready, they also have to balance work and training schedules, court visits, and family obligations [8]. Therefore, it is of importance to implement assessment procedures into their training schedule whenever possible. This can often be attained by choosing appropriate tests such as the 300-y shuttle run, 2.4 km Cooper test, and Illinois agility test.

  1. Marins, E.F.; David, G.B.; Del Vecchio, F.B. Characterization of the Physical Fitness of Police Officers: A Systematic Review. J Strength Cond Res 2019, doi:10.1519/JSC.0000000000003177.
  2. Schaal, M.; Ransdell, L.B.; Simonson, S.R.; Gao, Y. Physiologic Performance Test Differences in Female Volleyball Athletes by Competition Level and Player Position. J Strength Condi Res 2013, 27, 1841–1850, doi:10.1519/JSC.0b013e31827361c4.
  3. Sporiš, G.; Vučetić, V.; Milanović, L.; Milanović, Z.; Krespi, M.; Krakan, I. A Comparison Anaerobic Endurance Capacity in Elite Soccer, Handball and Basketball Players. Kinesiology 2014, 46, 52–59.
  4. Hachana, Y.; Chaabène, H.; Nabli, M.A.; Attia, A.; Moualhi, J.; Farhat, N.; Elloumi, M. Test-Retest Reliability, Criterion-Related Validity, and Minimal Detectable Change of the Illinois Agility Test in Male Team Sport Athletes. J Strength Cond Res 2013, 27, 2752–2759, doi:10.1519/JSC.0b013e3182890ac3.
  5. Streetman, A.; Paspalj, D.; Zlojutro, N.; Božić, D.; Dawes, J.J.; Kukić, F. Association of Shorter and Longer Distance Sprint Running to Change of Direction Speed in Police Students. NBP. Nauka, bezbednost, policija 2022, 27, 5–13, doi:10.5937/nabepo27-36289.
  6. Jones, A. Test and Measurment: 300-Yard Shuttle Run. Strength Cond J 1991, 13, 56–60.
  7. Bandyopadhyay, A. Validity of Cooper’s 12-Minute Run Test for Estimation of Maximum Oxygen Uptake in Male University Students. Biol Sport 2015, 32, 59–63, doi:10.5604/20831862.1127283.
  8. Lentine, T.; Johnson, Q.; Lockie, R.; Joyce, J.; Orr, R.; Dawes, J. Occupational Challenges to the Development and Maintenance of Physical Fitness Within Law Enforcement Officers. Strength Cond J2021, 43, 115–115, doi:10.1519/SSC.0000000000000679.

Materials and Methods

2.1. Study design and procedures

Do men and women do the same? Same tests?
No previous studies cited.

Response: Yes, women and men usually do the same tests when athletic performance is evaluated or when job-specific tests are performed. They may perform differently, but their physiological potentials are tested the same way. Also, while their performance may differ from each other, this study examined changes from baseline. Both men and women would be expected to have similar changes in their performance after the preceding tests. Our results show this clearly as they perform differently (as expected) but the trends are very similar. Although we noted differences in performance, this is not the focus of our research. We are showing that physical performance, although different, is similarly affected in both sexes (i.e., in homo sapiense) as physiological and biological processes are the same/similar regardless of sexes.

2.2. Participants

Males and females cannot be mixed.

A sample consisted of 50 students (42% female). The main sample characteristics 86 were age = 19.7 ± 0.5 years, body height = 177 ± 8 cm, body weight 74.7 ± 12.0 kg, and body mass index = 23.6 ± 2.6 kg/m2.

Response: We added separate main characteristics for male and female participants.

Need to indicate at which point in the training sessions the tests were performed.

Response: This is added in 2.3. Settings: “During the testing sessions, they did not have any other training activities”

Include for all tests and assessments the reference authors and their validity and reliability in measuring what they say they measure. This will be of great help to young researchers.

Response: We added the references for validity and reliability of the 300-y shuttle run and Cooper running test, while for the Illinois agility we already showed reliability on our sample.

Results They are adequately presented.

Response: Thank you.

Discussion In the discussion all statements should have previous research supporting or contradicting the results obtained. This should be reviewed. Many citations are missing.

Response: Dear Reviewer, we highly appreciate your concern, and we tend to agree that statements may need to be supported by citation. We highly value the effort of all researchers and we would happily cite any study that previously defined our statements. However, we would also like to point out that our study provides a clear quantitative determination of effects that have not been provided to date.    

The first paragraph discusses our results, without comparing them to any other study so we did not include any citation as these statements are based on our results rather than on previous studies.

The second paragraph puts our results into the perspective of the tactical population based on previous studies and cites 4 studies. True, these studies did not investigate the direct effects of running activity on CODS performance. However, they do provide a solid background to discuss the possible implications of our results. Note that our study is the first to our knowledge that investigated the effects of running with different intensities on subsequent CODS performance. Therefore, we could not cite other studies on this topic, and we believe that our study will serve as a reference for future studies.

The third paragraph discusses our results in the context of sports, where we cited 5 studies. These studies, like in the tactical population, did not investigate the direct effects of preceding running activity on subsequent CODS performance. We did not find studies including athletes that investigate these very important effects. Thus, our results reflect in statements that may not have references to cite but rather provide the reference for future studies. In addition, and not less important, our results are very practical and utilizable in sports and police training.

Conclusions They are clear.

Response: Thank you.

References Some minor flaws. Numbers together without separation. Use of inappropriate formatting in the year, number and pages.

Response: Thank you for the suggestion. We have carefully checked our references including the formatting for the year, number and pages. Due to the full justification format for the journal for the references, some spacing appears closer together than others.

Reviewer 2 Report

It is a study proposal to verify the effects of different performance tests in the change of direction among police students.

The study is poorly conducted. There is no solid scientific basis to confirm the proposal. The authors confuse terms of exercise physiology and perform inappropriate measurements. The tests used in the study have the same outcome, the estimate of maximum aerobic power. However, concepts such as lactic anaerobic running do not make any sense, especially in relation to the general objective of the study.

Authors should be clear on what they want to compare and review the literature, particularly the context of exercise physiology and terminology within exercise biochemistry. In addition, they must understand more about the tests performed and their purposes.

Unfortunately, this study does not meet the minimum characteristics to be accepted in any journal.

Author Response

Comments and Suggestions for Authors

It is a study proposal to verify the effects of different performance tests in the change of direction among police students.

The study is poorly conducted. There is no solid scientific basis to confirm the proposal. The authors confuse terms of exercise physiology and perform inappropriate measurements. The tests used in the study have the same outcome, the estimate of maximum aerobic power. However, concepts such as lactic anaerobic running do not make any sense, especially in relation to the general objective of the study.

Response: Although we appreciate the reviewer’s concern about having the same outcomes, respectfully, we must disagree that the 300-y shuttle runs and 2.4km Cooper test provide the same outcomes (maximum aerobic power). They both certainly do provide very high aerobic outputs, just the 300-y shuttle run also produces a very high accumulation of blood lactates. Not to mention that Illinois Agility Test does not provide the outcomes the reviewer suggests. When subsequent maximal activity needs to be performed immediately afterwards, it is not the same if participants’ blood lactate level is very high (violated blood’s pH) or not so high. In practice, this makes a big difference, otherwise, athletes would not need rest but rather would perform high-intensity activities all the time. This is why we call it lactic running. Maybe it is not 100% accurate physiologically, but it certainly is in practice. However, we made some adjustments in terminology for better clarity. Note also that the purpose of this study is (among others) to provide useful information for the improvement of exercise implementation in the field and to be easily understandable and applicable to strength and conditioning specialists.

Authors should be clear on what they want to compare and review the literature, particularly the context of exercise physiology and terminology within exercise biochemistry. In addition, they must understand more about the tests performed and their purposes.

Response: Thank you for this critique, however, we are not sure that we understand what you meant to say here. These are non-actionable general statements. To be frank, we think this comment is totally unprofessional and has no intention of improving the manuscript. Kindly elaborate on each statement so we understand the suggestion. It is impossible to act on these suggestions nor to refute them with any meaningful argument. The adjustments we made in terminology regarding the previous comment may accommodate this.

Unfortunately, this study does not meet the minimum characteristics to be accepted in any journal.

Response: We appreciate your opinion, it would not be our first study to get rejected. However, with a summed track record of all authors of this study (over 350 publications in peer-review Q1-Q4 journals) and practical everyday experience in applying exercise science in various populations, we believe this statement is rather subjective and not particularly supported by science and practice.

Reviewer 3 Report

Dear Authors

I reviewed # biology-1716916 entitled "Effects of anaerobic lactic and aerobic running on change of direction speed performance among police students" which you submitted to the biology. I would like to inform you of the consideration of your manuscript as follows.

Minor points

1.All statistic symbols 'p' should be changed to italics.

2.In general, it is recommended that the study design and procedures in line 67 be moved behind the study participants.

3.It is better to put the IRB number in line 94.

Major points

This study shows the result of a lot of effort. And it is logically well documented. However, it is thought that correction is necessary because there is an error in the statistical processing part.

1. In line 169, a detailed explanation of why the sample size was set to 50 is necessary. Please explain according to the sample size formula.

2. In statistical analysis, after considering the normal distribution, it is necessary to decide whether to analyze the data in a parametric or non-parametric method.

Sincerely,

Author Response

Comments and Suggestions for Authors

Dear Authors

I reviewed # biology-1716916 entitled "Effects of anaerobic lactic and aerobic running on change of direction speed performance among police students" which you submitted to the biology. I would like to inform you of the consideration of your manuscript as follows.

Response: Thank you for your effort and time.

Minor points

1. All statistic symbols 'p' should be changed to italics.

Response: We Italicized p values

2. In general, it is recommended that the study design and procedures in line 67 be moved behind the study participants.

Response: Thank you for the suggestion. However, respectfully, we followed the Biology journal template and this is how it should be there according to their guidelines. Of course, if needed, we do not mind shifting the study design and procedures behind the Participants.

3. It is better to put the IRB number in line 94.

Response: Moved to number 94.

 Major points

This study shows the result of a lot of effort. And it is logically well documented. However, it is thought that correction is necessary because there is an error in the statistical processing part.

1. In line 169, a detailed explanation of why the sample size was set to 50 is necessary. Please explain according to the sample size formula.

Response: We expanded the participants and statistics section with an explanation of the sample size.

This is in the participants section: “Consulting the G*power analysis (see Statistics section) for the minimal sample size, we initially started the study with 57 participants, but three female and four male students dropped out during the study. One student had to leave the campus and miss several testing sessions due to a death in the family, four students missed several testing sessions due to sickness, and two students left the study due to lack of motivation”

This is added at the end of the statistics: “We used G*Power to determine that the minimum sample sufficient to detect large effect size. For the independent sample t-test, 52 participants (26 in each group) were recommended, for the paired sample t-test 15 participants were recommended, and for the repeated measures analysis of variance 10 participants were recommended.”

2. In statistical analysis, after considering the normal distribution, it is necessary to decide whether to analyze the data in a parametric or non-parametric method.

Response: Fair point, thank you for the suggestion. We expanded the Statistics section with explanations regarding the parametric and non-parametric tests.

Round 2

Reviewer 1 Report

Dear authors, thank you for making the relevant changes.

Author Response

Thank you for your time and effort to help us make our article better.

Reviewer 2 Report

Dear authors,
First, I apologize if my initial opinion seemed aggressive and unprofessional. My sincere apologies. Here, I will try to be briefly clear and assertive in my propositions about the study, but initially I will make some observations:

1) As you informed that you are the authors of hundreds of studies, you certainly reviewed other manuscripts of your peers. Therefore, understand that the reviewer is also an expert on the subject. So, the first mistake in this speech is the use of the authority argument that you used to try to reduce my work. Therefore, as a suggestion, a researcher should avoid using their authority (eg over 350 publications in peer-review Q1-Q4 journals).

The other comments I will not comment. I go to the point of the study submitted for evaluation, which is the object of interest:

About terminology and biochemistry The authors consider the Shuttle-Run test to be an "anaerobic" running.

The first question is: What is an anaerobic run?

Although this is a rhetorical question, let's clarify a few points. The Shuttle-Run test is one of the main tests used to estimate cardiorespiratory fitness (Castro-Piñero et al., 2010). These are 20m races with an increase in speed until exhaustion or non-compliance with the premises. That is, as the speed (imposed load) increases, the oxygen consumption will also increase proportionally (see Di Prampero studies). In addition, it is expected that the lactate concentration will increase, due to the characteristic of the movement. However, testing remains progressive and incremental. In the end, we can estimate the maximum aerobic speed as well as the VO2max. However, it is worth remembering that during the increase in imposed load, some variables may respond disproportionately (eg ventilation and lactate), and some call this point the anaerobic threshold. Therefore, efforts above the threshold should not be characterized as anaerobic. From the physiological point of view (basic physiology), there is an increase in the contribution of anaerobic systems (glycolytic pathway), however, the predominance of effort will still be aerobic. Therefore, the shuttle-run is not an anaerobic test, nor is it intended to verify anaerobic power. For this, there are other tests, such as RAST. So, I believe, until the authors prove otherwise, that the shuttle-run should not be referred to as an anaerobic run. Although, I believe I don't need to elaborate further, this terminological and biochemical confusion prevents the justification of the study in question from being plausible. I ask the authors, therefore, to understand this point and change the objective. One suggestion is to compare different methods (shuttle vs 2.4km) at different intensities on the agility test.

The use of references of (4mm of [LAC]) should be cautious, since the authors only measured the [LAC] after 3min of maximal shuttle-run test.

Why wasn't [LAC] measured after the 2.4km Cooper test?

minor

Figure 1. Needs to be edited to the point where all items are readable. At the moment it is unclear.

Author Response

Dear authors,

First, I apologize if my initial opinion seemed aggressive and unprofessional. My sincere apologies. Here, I will try to be briefly clear and assertive in my propositions about the study, but initially I will make some observations

Response: Dear Reviewer, we did not take your review offensively, and we think it still improved our manuscript. We just think that the way you communicated it was not professional, and it could have been more actionable. However, we appreciate your explanation.  

1) As you informed that you are the authors of hundreds of studies, you certainly reviewed other manuscripts of your peers. Therefore, understand that the reviewer is also an expert on the subject. So, the first mistake in this speech is the use of the authority argument that you used to try to reduce my work. Therefore, as a suggestion, a researcher should avoid using their authority (eg over 350 publications in peer-review Q1-Q4 journals).

Response: We too sincerely apologize if you felt in any way that we tried using the authority to reduce your work. That was not our intention at all. We certainly appreciate the time and effort you put into reviewing our manuscript. Thank you!

The other comments I will not comment. I go to the point of the study submitted for evaluation, which is the object of interest:

About terminology and biochemistry The authors consider the Shuttle-Run test to be an "anaerobic" running.

The first question is: What is an anaerobic run?

Although this is a rhetorical question, let's clarify a few points. The Shuttle-Run test is one of the main tests used to estimate cardiorespiratory fitness (Castro-Piñero et al., 2010). These are 20m races with an increase in speed until exhaustion or non-compliance with the premises. That is, as the speed (imposed load) increases, the oxygen consumption will also increase proportionally (see Di Prampero studies). In addition, it is expected that the lactate concentration will increase, due to the characteristic of the movement. However, testing remains progressive and incremental. In the end, we can estimate the maximum aerobic speed as well as the VO2max. However, it is worth remembering that during the increase in imposed load, some variables may respond disproportionately (eg ventilation and lactate), and some call this point the anaerobic threshold. Therefore, efforts above the threshold should not be characterized as anaerobic. From the physiological point of view (basic physiology), there is an increase in the contribution of anaerobic systems (glycolytic pathway), however, the predominance of effort will still be aerobic. Therefore, the shuttle-run is not an anaerobic test, nor is it intended to verify anaerobic power. For this, there are other tests, such as RAST. So, I believe, until the authors prove otherwise, that the shuttle-run should not be referred to as an anaerobic run. Although, I believe I don't need to elaborate further, this terminological and biochemical confusion prevents the justification of the study in question from being plausible. I ask the authors, therefore, to understand this point and change the objective. One suggestion is to compare different methods (shuttle vs 2.4km) at different intensities on the agility test.

Response: We agree with the reviewer that the 20m shuttle-run test is an aerobic test. There is no doubt about that, and the review study by Castro-Piñero et al. (2010) nicely explains this. However, we did not use this test in our study. We used 300-yard shuttle-run test where the running speed is not gradually increased, but participants run the whole test maximally. They were instructed to run as fast as possible and verbally encouraged during the test “to use all they have got.” It is rather an all-out field test where participants often reach failure at the end or near the end due to the large accumulation of blood lactates and the inability of the aerobic system and other buffer systems to metabolize waste products of anaerobic lactic activity.

The use of references of (4mm of [LAC]) should be cautious, since the authors only measured the [LAC] after 3min of maximal shuttle-run test. Why wasn't [LAC] measured after the 2.4km Cooper test?

Response: We measured lactate level only after the 300y shuttle run test because that is an anaerobic lactic test, and we just wanted to make sure participants did the text maximally. The 2.4km Cooper test is an aerobic test, and we did not check the blood lactate level, but we did check their heart rate. If we used a 20m shuttle run test, an aerobic test as you already pointed out, we would not collect the blood lactates but heart rate.

minor
Figure 1. Needs to be edited to the point where all items are readable. At the moment it is unclear.

Response: We edited Figure 1 for better clarity. 

Round 3

Reviewer 2 Report

Dear authors,

Thanks for your responses. 

We agree with you. Congrats